# Task-Adaptive Tokenization: Enhancing Long-Form Text Generation Efficacy in Mental Health and Beyond

**Siyang Liu**🐱     **Naihao Deng**🐱     **Sahand Sabour**🚀
**Yilin Jia**🐱     **Minlie Huang**🚀     **Rada Mihalcea**🐱

🐱Language and Information Technologies Lab (LIT),
Department of Computer Science and Engineering, University of Michigan, Ann Arbor
🚀The CoAI group, Tsinghua University, Beijing
{lsiyang, mihalcea}@umich.edu, liusyang641@gmail.com

## Abstract

We propose task-adaptive tokenization[1] as a way to adapt the generation pipeline to the specifics of a downstream task and enhance long-form generation in mental health. Inspired by insights from cognitive science, our task-adaptive tokenizer samples variable segmentations from multiple outcomes, with sampling probabilities optimized based on task-specific data. We introduce a strategy for building a specialized vocabulary and introduce a vocabulary merging protocol that allows for the integration of task-specific tokens into the pre-trained model's tokenization step. Through extensive experiments on psychological question-answering tasks in both Chinese and English, we find that our task-adaptive tokenization approach brings a significant improvement in generation performance while using up to 60% fewer tokens. Preliminary experiments point to promising results when using our tokenization approach with very large language models.

## 1 Introduction

During a time when mental health support is quickly growing (Hock et al., 2012; WHO, 2020), text generation techniques have been identified as potentially useful tools to assist mental health professionals (MHP) and provide mental health support to those who share their struggles online (Demasi et al., 2020; Liu et al., 2021a; Sabour et al., 2022). With the help of such generation tools, counselors and social workers alike can increase their work efficiency and offer timely and expert feedback to those in need, especially for those in under-resourced areas where access to mental health services is challenging (Patel et al., 2011; Brenman et al., 2014; Shen et al., 2022a).

The task of Psychological Question-Answering (PsyQA) is to generate a supportive response to a

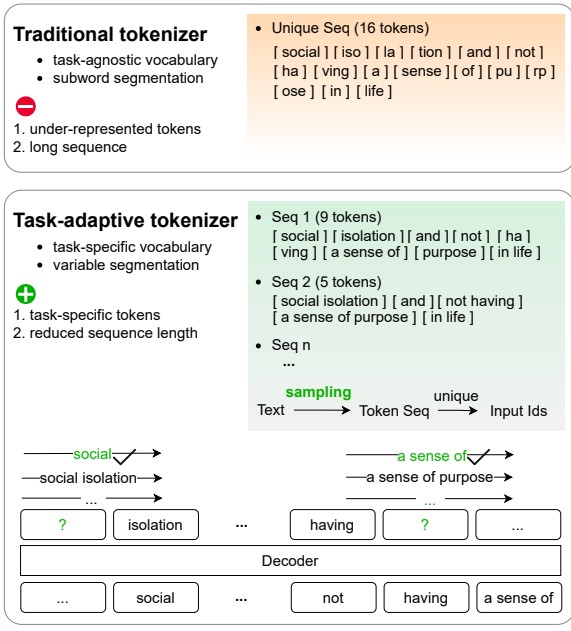

E.g., "**Social isolation and not having a sense of purpose** in life have been linked to mood disorders"

Figure 1: A brief comparison between task-adaptive and traditional tokenizer. With task-adaptive tokenization, the same phrase appearing in different training instances may result in different tokenizations. For instance, Seq 1 and 2 illustrate two different token sequences of the same input using a task-adaptive tokenizer.

help-seeking post, with an important requirement that the language used in the response aligns with the style used by MHPs (Sun et al., 2021; Welch et al., 2020). The task poses significant challenges due to its domain-specific terminology, text structure, and language style, which are often underrepresented in the pre-training data. Recent research has demonstrated that the linguistic patterns of MHPs align with established counseling principles and behaviors (Lahnala et al., 2021), thus exacerbating the difficulty of the task as pre-training data is often sourced from laypeople (Kalla and Smith, 2023). Despite recent progress in large-scale language models (LLMs) (Touvron et al., 2023; Cui et al., 2023; Hu et al., 2022), we found that few-

---

[1]Our work is available at github.com/MichiganNLP/task-adaptive_tokenization.

shot abilities of these models lag behind in mental health applications due to this misalignment with the MHP linguistic behaviors and style (see Appx A). This requires us to tune models to the downstream domain. However, much of the current work focused on fine-tuning (Shen et al., 2020; Ji et al., 2022) still yields unsatisfying results due to the scarcity of professional data, which is often insufficient to capture all the task-specific intricacies.

In this paper, we introduce task-adaptive tokenization as a strategy to adapt a model to a downstream task. Rather than focusing the adaptation effort on post-training fine-tuning, which has been the typical approach used in recent work (Shen et al., 2020, 2022b; Sun et al., 2021; Lee et al., 2021), we propose to make task-specific adjustments in the way the language is segmented during the tokenization stage (see our motivation elaborated in Sec 3). Thus, we propose task-adaptive tokenization, which is built specifically from the task data and equipped with variable text segmentation, yet can be seamlessly integrated into off-the-shelf language models. To illustrate the potential of task-adaptive tokenization, we show an example in Fig 1. As seen in this example, our tokenization process allows for the inclusion of domain-specific terminology such as [_social_isolation] and [_a_sense_of_purpose] in the vocabulary. A model trained with task-specific tokenization is now able to generate these tokens through learned preference, which we show can lead to significant performance improvements.

We confront three major challenges when designing tailored tokenization strategies. First, The creation of a task-specific tokenization vocabulary must be performed through an automatic process due to the labor-intensive and time-consuming nature of manual selection. Integrating this task-specific vocabulary seamlessly with pre-trained models poses is challenging, and it requires techniques to fuse the task-specific vocabulary with the pre-trained vocabulary and fine-tune the resulting model accordingly. Lastly, we need to address the poor representation of newly added tokens in task-specific vocabularies that were not learned during the pre-training phase.

In this paper, we propose task-adaptive tokenization as a method for enhancing text generation in specialized tasks such as PsyQA. This paper makes three main contributions. (1) Building on insights from cognitive linguistics (Thorndyke,

1977; Wells, 1947), we advocate for using task-specific data and the developing variable segmentation for a downstream vocabulary as a pre-step for creating a task-adaptive tokenizer. (2) We construct a protocol for merging task-specific and pre-trained vocabularies, allowing for fine-tuning inputs to be sampled from multiple tokenization results. (3) We propose a simple yet effective initialization mechanism to alleviate the difficulty of learning representations for new tokens unseen during pre-training. Through thorough experiments on the PsyQA task in both Chinese and English, we demonstrate the significant improvements achieved by our task-adaptive tokenization approach. Notably, we achieve these enhancements while utilizing 60% fewer tokens compared to expressing equivalent content length. In addition, we show that our tokenization brings significant improvement in 7B LLaMA models, which suggests that our method is effective regardless of the model size and can unlock additional performance despite the booming era of LLMs.

## 2 The PsyQA Task

The goal of the PsyQA task is to generate a supportive response to the help-seeker via responding to their post, where an essential requirement is to imitate the use of language that is characterized as professional by previous work (Sun et al., 2021; Welch et al., 2020). Figure 3 shows an example of a question-answer pair in this dataset. Posts and responses are often extensive, with help-seekers providing detailed accounts of their experiences, and those offering assistance providing comprehensive views, including emotional comfort, in-depth analysis, and various suggestions.

The formal definition of the task is as follows: given a question $Q$, a description $D$, and keywords $K$, let context $C$ denote an integral description of Q, D, and K; $\mathbf{c} = (c_1, c_2, ..., c_m)$ is the sequence by segmenting the context $C$. We aim to generate a response $R$ (a sequence of $\mathbf{r} = (r_1, r_2, ..., r_n)$) corresponding to the context $C$.

## 3 Motivation

Our motivation builds on arguments stemming from cognitive science, where (1) a clear distinction is being made between the vocabulary used to interpret language versus the vocabulary used for language production; and (2) there is evidence for increased efficiency in speech behavior stemming

from individual segmentation granularities. These arguments are further expanded by optimization and efficiency goals, which are better achieved in the presence of flexible segmentation.

## 3.1 Receptive vs Productive Vocabulary

Within cognitive linguistic research, a clear distinction is being made between "receptive" and "productive" vocabulary (Bogaards and Laufer, 2004; Stæhr, 2008) – the former referring to words comprehended while reading, and the latter to words actively used in writing or speaking. A strong productive vocabulary has a direct impact on writing quality (Engber, 1995; Fajrina et al., 2021) and is essential for effective communication and precise articulation, particularly in technical fields where specialized terminology is common (Maskor et al., 2016; Faraj, 2015). We, therefore, hypothesize that while creating a large-scale vocabulary is essential for training language models (i.e., the "receptive" vocabulary), generation tasks require more emphasis on designing and leveraging task-related vocabulary (i.e., the "productive" vocabulary).

To illustrate this gap in practice, considering the PsyQA task as described earlier, a typical optimization objective used for fine-tuning would be

$$\theta_{MLE} = \underset{\theta}{\arg\max}\, L(\theta), \qquad (1)$$

where

$$L(\theta) = \sum_{s=1}^{|D|} log P(r^{(s)}|c^{(s)}; \theta)$$

Here, $c$ and $r$ are sequences of tokens, i.e., the segmentations of the input context $C$ and the response $R$. The input of the function starts from $c$ and $r$ instead of the original texts $C$ and $R$, due to the common practice of using a vocabulary that determines the mapping relationship between texts and tokens. Thus, vocabulary construction is not necessarily considered in the process of optimization.

However, if we do not assume the vocabulary in this process, we obtain the log-likelihood:

$$L(\theta) = \sum_{s=1}^{|D|} \underset{c \sim p(c|C^{(s)})}{E_{r \sim p(r|R^{(s)})}} log P(r|c; \theta), \qquad (2)$$

where

$$\{< C^{(s)}, R^{(s)} >\}_{s=i}^{|D|} \overset{\text{tokenizer}}{=} \{< c^{(s)}, r^{(s)} >\}_{s=i}^{|D|}$$

As seen in Equation (2), different segmentations of a text can influence the entropy of the training corpus and thus can influence the model's performance.

In practice, researchers often blindly adopt a pre-existing vocabulary without considering the potential distribution inconsistency between the train data (typically used to generate the vocabulary) and the data of the downstream task, which can hinder the downstream performance (Liu et al., 2021c). For example, the data on which the word-piece model is trained to obtain the BERT vocabulary originates from Google's Neural Machine Translation benchmark (Wu et al., 2016). The composition of this vocabulary is designed for machine translation, which may not be ideal for PsyQA or for other tasks, according to Equation (2). This additionally supports our argument that a custom vocabulary informed by the task at hand is needed to achieve the best optimization potential.

## 3.2 Efficient Speaker Behavior

Empirical studies have demonstrated that during language production, individuals tend to pause at various syntactic constituents, such as sentence boundaries or between clauses (Abney, 1992; Gee and Grosjean, 1984; Torrance et al., 2007). This phenomenon, referred to as "pause behavior," has been a popular research topic in cognitive linguistics (Thorndyke, 1977; Wells, 1947). A possible explanation for this behavior is the fact that different individuals produce texts at various granularities, from single letters and words to phrases and even entire sentences. When certain expressions are regularly used, they are stored as whole units in our memories, thereby reducing the cognitive load for future usage.

Building on this argument, we hypothesize that implementing a similar strategy in text generation can equally lead to more efficient behavior. Similar to human speakers, with variable segmentation, we can accommodate tokens at the sub-word level to address the rare word problem (Sennrich et al., 2015; Wu et al., 2016) while also including larger granularity units such as phrases and clauses.

This argument is further supported by previous work that has demonstrated that a fine-grained segmentation, despite its flexibility, can lead to increased computational cost and degradation in token representation (Zouhar et al., 2023; Liu et al., 2021c; Yu et al., 2021; Demeter et al., 2020; Wang

et al., 2023). For instance, recent large language models such as GPT3 may require roughly two tokens for the representation of an eight-letter length word[2]. This fragmentation also leads to tokens such as [ soci ], [ _al ] or [ _iso ], which are often shared by many words and lead to underrepresentation, preventing the fine-tuned model to better learning the compositionality of generation (Liu et al., 2021c; Dou et al., 2021; Yu et al., 2021). Instead, if we could allow for more cohesive information to be represented in a token, including for instance task-specific tokens such as [ social_isolation ], we could potentially reduce the computational cost and achieve stronger token representation.

## 4 Task-adaptive Tokenization

Our approach to task-adaptive tokenization consists of three main steps:

1. **Task Vocabulary Construction:** First, we compile a task-specific vocabulary (Sec 4.1) by leveraging a subword regularization algorithm.
2. **Vocabulary Merging:** Next, we merge the task-specific vocabulary with the original vocabulary from pre-trained models (Sec 4.2).
3. **Token Mapping:** Finally, we create new token embeddings by mapping the new token to the sub-words in the pre-trained vocabulary and averaging the sub-word embeddings (Sec 4.3).

### 4.1 Task Vocabulary Construction

To construct a task-specific vocabulary that allows for variable segmentation, as described in Section 3, we use subword regularization (Kudo, 2018). Subword regularization optimizes the likelihood of the training corpus on all possible text segmentations and produces a vocabulary that consists of a set of tokens and their corresponding log-likelihood scores. Specifically, this algorithm leverages a regularization coefficient to increase the sampling probability of low-score segmentations during training to learn the representations of various segmentations. This allows for sampling of a certain segmentation among various possible segmentations based on the score of the text being segmented. To adapt the original algorithm to our setting, we use task-specific data (i.e., all the response sentences from the QA pairs in the PsyQA task) to train a unigram language model. In addition, contrary to the original algorithm, we do not

[2]platform.openai.com/playground

split sentences into words, as we want to include segmentations of various granularities.

To illustrate, consider the sentence "a sense of purpose in life" as an example. The original model segments it into subwords as follows:

- [ _a ] [ _sen ] [ se ] [ _of ] [ _purpose ] [ _in ] [ _life ]
- [ _a ] [ _sense ] [ _of ] [ _pur ] [ pose ] [ _in ] [ _life ]

With our modification, the model is also able to produce the following additional segmentations:

- [ _a_sense ] [ _of ] [ _purpose ] [ _in_life ]
- [ _a_sense_of_purpose_in_life ]

Alg 1 shows the task vocabulary construction process, where underline text indicates our modifications to the original algorithm.

---

**Algorithm 1:** Task-specific Vocabulary Construction

**Data:** $D = \{\lambda s : s \in \text{task dataset}\}$
**Result:** $V = \{(t_1, l_1), ..., (t_N, l_N)\}$, a vocabulary with size $N$, where $t_i$ is the $i$th token and $l_i$ is the corresponding score.

1 $D \longrightarrow P = \{\lambda p : p \in \text{pieces cut at any length}\}$ ;

2 $P \longrightarrow V_{big} = \{t_1, ..., t_M\}$, where $M >> N$;

3 Unigram-model optimizes $\mathcal{L} = \sum_s^{|D|} log(\sum_{\mathbf{t} \in S(t^s)} P(\mathbf{t}))$ , where $S(t)$ is all possible segmentation for $s$;

4 Compute the $loss_i$ of each $t_i$ in $V_{big}$;

5 Sort $(t_1, loss_1), ...(t_M, loss_M)$;

6 $\{(t_1, loss_1), ...(t_M, loss_M)\} \xrightarrow{truncate} V$.

---

The process followed in each step in the algorithm is as follows:

1. Divide all sentences $s \in D$ into various granularity pieces;
2. Choose the most frequent pieces and the union of characters up to a big seed vocabulary $V_{big} = \{t_1, ..., t_M\}$ where $M >> N$;
3. Build a unigram model on $D$. The probability of a possible segmentation $\{t_1, ..., t_K\}$ on $s$ is $P(\mathbf{t}) = \prod_1^K p(t_i)$. The most possible segmentation is optimized by $\mathbf{t}^* = argmax_{\mathbf{t} \in S(t)} P(\mathbf{t})$, where $S(t)$ is all possible segmentations for $s$. Apply EM algorithm to unigram model with the objective function $\mathcal{L} = \sum_s^{|D|} log(\sum_{\mathbf{t} \in S(t^s)} P(\mathbf{t}))$;

4. Compute the $loss_i$ of each $t_i$ in $V_{big}$ , where $loss_i$ represents how the likelihood $\mathcal{L}$ has reduced when the piece $t_i$ is removed from the current vocabulary;

5. Sort the tuples $(t_1, loss_1), ...(t_M, loss_M)$ in the descending order;

6. Keep the union of characters and the pieces with the highest $loss$ score until it satisfies the target vocabulary size $N$; Get the final vocabulary $V = \{(t_1, l_1), ..., (t_N, l_N)\}$, where $l_i$ is the log probability of $t_i$ in the unigram model.

## 4.2 Vocabulary Merging

After creating a task-specific vocabulary, an important challenge is how to incorporate this "alien" vocabulary in the pre-trained models, as such models already have their own vocabulary and corresponding embeddings. To this end, we build a protocol for merging a task-specific vocabulary with a pre-existing vocabulary (referred to as original vocabulary), as shown in Fig 2.

The design of such a merging protocol considers two aspects. First, to inherit the embedding matrix of the pre-trained models, the order of all tokens from the original vocabulary is maintained in the merged vocabulary. In this way, new representations could be added to the embedding matrix by extending the rows of the original one (Rule 4 in Fig 2). Second, special tokens and the tokens from the original vocabulary that are constructed based on unique segmentation algorithms (e.g., WordPiece and BPE) do not have a score for sampling. Thus, we have to assign them an appropriate score based on our guidelines (Rules 1, 2, and 3 in Fig 2). We assign $-bigScore * \frac{len(token)+1}{len(token)}$ to those tokens qualifying for Rule 2, where the $bigScore$ ought to be lower than the lowest score among task-specific tokens, to ensure task-specific tokens have a higher priority to be sampled; meanwhile, longer tokens will receive a bigger score than their sub-string tokens, following the BPE/WordPiece design that prioritizes longer segments. Please see token statistics for a merged vocabulary in Appendix B.

## 4.3 Token Mapping

To address the possible difficulties of representation learning for new tokens that are never seen during pre-training, we propose a simple but effective initialization mechanism to alleviate the problem. For each new token, we acquire its subword

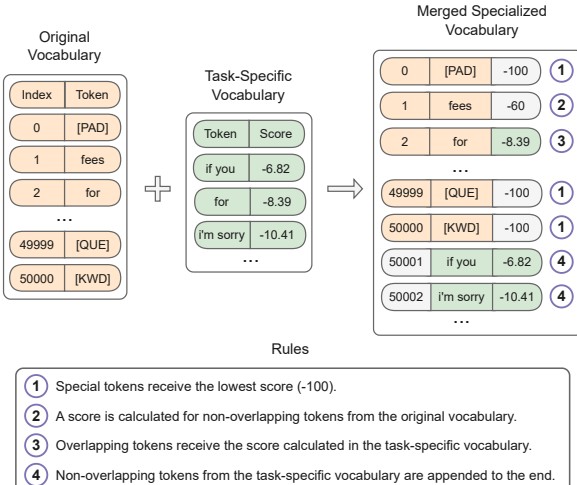

**Rules**

① Special tokens receive the lowest score (-100).

② A score is calculated for non-overlapping tokens from the original vocabulary.

③ Overlapping tokens receive the score calculated in the task-specific vocabulary.

④ Non-overlapping tokens from the task-specific vocabulary are appended to the end.

Figure 2: An overview of the merging protocol.

embeddings by using the primary pre-trained tokenizer to segment the new token into a set of subwords. The initial embedding of new tokens is set as the average of its subwords' embeddings from the original embedding matrix. For overlapping tokens, we leverage the existing embedding representations.

## 5 Experiments

We assess the effectiveness and efficiency of our task-adaptive tokenization on the PsyQA task through several automatic and manual evaluations.

### 5.1 Datasets

**CN PsyQA** is a Chinese dataset of psychological QA support, where the answers are provided by well-trained volunteers and counselors (Sun et al., 2021). This dataset contains 22,346 questions and 56,063 answers, where each question relates to one or more ground-truth responses.

**MHP Reddit** is an English dataset of Post-Response pairs crawled from Reddit, where responders self-identify as mental health professionals in their profiles (Lahnala et al., 2021). Compared to CN PsyQA, this dataset is smaller in size (9,501 QA pairs), and the responses are relatively shorter.

### 5.2 Compared Methods

We compare our task-adaptive tokenizer with the tokenizers from off-the-shelf pre-trained models. We acknowledge that the implementation details of a tokenizer are bound to the pre-trained model type and corpora. We provide details of the tokenizers we compare against in Tab 1, includ-

ing the associated model type, vocabulary size, and the segmentation algorithm. For brevity, we use the notations $[LLM_{base}]$ and $[LLM_{TaT}]$ under PsyQa or MHP datasets to represent the corresponding base and task-adaptive tokenizers for a model named $LLM$, respectively. Accordingly, we append the *w/o mapping* to the notations to indicate whether the mapping mechanism is used.

| Dataset | Model source | Tokenizer | Vocab size |
|---------|--------------|-----------|------------|
| CN PsyQA | gpt2-chinese-cluecorpussmall[3] | WordPiece | 21128 |
| | bart-base-chinese[4] | WordPiece | 51271 |
| | chinese-LLaMA-plus-7B [5] | BPE-dropout | 49953 |
| MHP Reddit | gpt2[6] | BPE | 50257 |
| | bart-base[7] | BPE | 50265 |

Table 1: Tokenizers used in our comparative evaluations

## 5.3 Experiment Details

As our backbone models, we adopt the small versions of GPT2 (Radford et al., 2019) and BART (Lewis et al., 2019) to analyze the generative improvements in both decoder-only and encoder-decoder models. Additionally, to study the effectiveness of our approach on larger language models, we include a 7B LLaMA model in our experiments (Touvron et al., 2023). However, due to limited resources, we were only able to conduct experiments on the Chinese version of LLaMA (Cui et al., 2023). For training, we create 8:1:1 splits of the datasets (section 5.1) and fine-tune our small backbone models (i.e., GPT2 and BART) while only training the LoRA adapter (Hu et al., 2022) and the input & output embeddings of LLaMA. Additional details for training and generation are included in Appx C.

## 5.4 Automatic Evaluation

**Effectiveness.** We use the HuggingFace evaluation tools (Wolf et al., 2019) and report Bleu-1 (**B-1**), Bleu-3 (**B-3**), average score of Bleu-1, 2, 3 and 4 (**Bleu**) (Papineni et al., 2002) and **RougeL** (Lin, 2004) for the generation on the test set. Although such reference-based metrics may not be sufficient for open-ended generation tasks (Zhang* et al.,

---

[3]huggingface.co/uer/gpt2-chinese-cluecorpussmall
[4]huggingface.co/fnlp/bart-base-chinese
[5]huggingface.co/ziqingyang/chinese-llama-plus-lora-7b
[6]huggingface.co/gpt2
[7]huggingface.co/facebook/bart-base

---

| CN PsyQA | | | | | | |
|---------|------|------|------|------|------|------|
| Setting | Bleu | +pct | B-1 | B-3 | R-L | +pct |
| gpt$_{ft}$+s* | 20.1 | | | | - | |
| GPT2$_{base}$ | 18.2 | - | 55.5 | 2.5 | 15.5 | - |
| GPT2$_{TaT}$ | 24.8† | +35.9% | 65.7† | 6.4† | 27.1† | +74.8% |
| +mapping | 25.0† | +37.1% | 66.3† | 6.6† | 22.1† | +42.1% |
| Bart$_{base}$ | 21.6 | - | 62.3 | 4.0 | 21.8 | - |
| Bart$_{TaT}$ | **26.2†** | +21.3% | **69.2†** | 6.7† | 27.2† | +24.8% |
| +mapping | 26.1† | +20.8% | 68.8† | **6.7†** | **27.2†** | +24.8% |

| MHP Reddit | | | | | | |
|-----------|------|------|------|------|------|------|
| Setting | Bleu | +pct | B-1 | B-2 | R-L | +pct |
| GPT2$_{base}$ | 3.7 | - | 14.0 | 0.6 | 5.7 | - |
| GPT2$_{TaT}$ | 3.6 | -2.7% | 13.0 | 1.3† | 8.1† | +42.1% |
| +mapping | **4.5** | +16.4% | **16.3†** | **1.6†** | **9.0†** | +57.9% |
| Bart$_{base}$ | 6.7 | - | 23.5 | 2.6 | 10.8 | - |
| Bart$_{TaT}$ | **7.6†** | +13.4% | **27.9†** | 2.5 | 10.1 | -6.5% |
| +mapping | 6.7 | +0.0% | 22.9 | **3.0** | **10.9** | +0.9% |

Table 2: Generation effectiveness. Bleu is calculated by averaging B-1,2,3,4, where B-n denotes the Bleu n-gram precision. R-L is RougeL score. +pct denotes the percentage of improving scores corresponding to Bleu and RougeL over the base. * indicates the sota results reported by Sun et al. (2021), who fine-tuned GPT2 with auxiliary support strategy information. † indicates a significant improvement over the base (p-value < 0.05).

2020), we report these results as a reference since they are commonly used by previous research (Gu et al., 2023; Cao and Wang, 2022; Yue et al., 2021). In addition, we believe that the multiple gold responses provided for each question in the CN PsyQa dataset alleviate the shortcoming of reference-based metrics to a degree. We leverage character-level comparisons for Chinese and word-level comparisons for English. To establish a fair comparison, we apply an NLTK word tokenizer (Bird et al., 2009) to all generated responses.

From Tab 2, the task-adaptive tokenizer consistently outperforms the baseline tokenizers, with a maximum increase of 37.1% on Bleu and 74.8% on RougeL. The results demonstrate two important insights. First, task-adaptive tokenization shows a larger increase on Bleu-3 than Bleu-1, indicating that variable segmentation can enhance the expression of task-specific phrases. Second, the increase in RougeL suggests a successful retrieval of task-specific expressions from the dataset.

However, since the results from the automatic evaluation do not indicate large improvements for the mapping mechanism on CN PsyQA, we turn

to human evaluations in Sec 5.5, and the results demonstrate that the mapping mechanism is important for the generation quality in human perception. To also gain insights into the weaker effectiveness of task-adaptive tokenization on Reddit MHP, in addition the human evaluation conducted to validate its effectiveness, in Sec 6 we extend our experimentation by creating a parallel English corpus, translated from CN PsyQA. This addition allows us to probe the underlying factors contributing to the limited improvement observed in Reddit MHP – whether the disparity in performance can be attributed to linguistic differences (English versus Chinese) or to disparities in data quality observed within the Reddit MHP dataset.

**Efficiency.** In order to assess generation efficiency, we employ various metrics on the test set, including the average number of centiseconds per generation (**#cSec**), the average number of tokens per generation (**#Tok**), response length (**Len**), generation length per token (**Len/#Tok**) and generation length per centisecond (**Len/#cSec**). To calculate response length, we consider token-agnostic measures, such as the number of characters for Chinese and the number of words after whitespace-splitting for English. The token utilization rate is then derived by dividing the number of tokens by the response length.

Tab 3 indicates that by using task-adaptive tokenization, models trained on Chinese and English datasets use significantly fewer tokens to represent more content. This enhancement is more apparent in Chinese, as its generation length per token is increased from 0.79 to 2.00, indicating a more than double time increase in token utilization rate. However, the results only show a significant improvement in generation speed for the Chinese dataset. We believe this occurs as responses in the MHP Reddit dataset are rather short and thus benefit less from using fewer tokens for generation, which is compensated by the increased time consumption from the expanded embedding layer.

**Vocabulary Size.** We also investigate the influence of different task-specific vocabulary sizes on generation quality (Appx D). The results indicate that an optimal size of vocabulary for merging may exist, but the difference may be subtle.

## 5.5 Human Evaluation

We recruit five native professionals for the human evaluation of the Chinese and English results, respectively. Prior to the evaluation, participants

| Setting | #cSec | #Tok | Len | Len/#Tok ↑ | Len/#cSec ↑ |
|---|---|---|---|---|---|
| **CN PsyQA** | | | | | |
| GPT2$_{base}$ | 5.7 | 440.2 | 365.9 | 0.8 | 64.2 |
| GPT2$_{TaT}$ + mapping | 3.6 | 190.3 | 382.9 | **2.0** | **106.4** |
| **MHP Reddit** | | | | | |
| GPT2$_{base}$ | 1.6 | 117.1 | 86.9 | 0.7 | **54.3** |
| GPT2$_{TaT}$ + mapping | 2.4 | 118.8 | 123.5 | **1.0** | 51.5 |

Table 3: Efficiency of generation. #cSec and #Tok denote the average number of centiseconds and tokens per generation on the test set respectively. Length denotes the average length of generated responses.

| Metric | M vs. B Win | M vs. B Lose | NM vs. B Win | NM vs. B Lose | M vs. NM Win | M vs. NM Lose |
|---|---|---|---|---|---|---|
| **CN PsyQA** | | | | | | |
| F | 31[†] | 15 | 18 | 24 | 36[†] | 11 |
| C | 37[†] | 9 | 19 | 19 | 36[†] | 10 |
| PE | 23 | 20 | 18 | 22 | 32[†] | 13 |
| **MHP Reddit** | | | | | | |
| F | 26 | 20 | 4 | 43 | 44[†] | 4 |
| C | 28 | 20 | 4 | 38 | 48[†] | 1 |
| PE | 30 | 18 | 6 | 39 | 45[†] | 3 |

Table 4: Human Evaluation. An explanation for abbreviations: M for GPT2$_{TaT}$ +mapping, B for GPT2$_{base}$, and NM for GPT2$_{TaT}$ w/o mapping; F for fluency, C for coherence, and PE for professional expression. Ties are not shown. [†] denotes a significant win (one sample sign test, p-value < 0.05).

were provided with ten QA pairs that were considered the "gold standard." They were instructed to familiarize themselves with the wording, structure, and language style of these gold responses, serving as a calibration exercise to assess the professional expression of a response and how closely it aligns with the standard responses. Each participant underwent ten rounds of evaluation under a guideline (see Fig 4). In each round, they were presented with a set of posts and corresponding response triplets, comprising the responses from GPT2$_{base}$, GPT2$_{TaT}$ w/o mapping, and GPT2$_{TaT}$ with mapping. The participants were then tasked with ranking the three responses based on three aspects: (1) **Fluency**: the response's fluency and readability, (2) **Coherence**: the responsiveness of the response to the post's content and its logical consistency, and (3) **Professional expression**: the proximity of the generated response to the standard responses in terms of wording, structure, and language style.

From the findings presented in Tab 4, the inclusion of a mapping mechanism is crucial for ensuring a robust token representation system, particularly when dealing with small-scale data (MHP Reddit). Without this mechanism, the generated responses exhibit a significant decline across three aspects, despite an increase in automatic evaluation scores. Moreover, our tokenization approach with the mapping mechanism outperforms the baseline on CN PsyQA in human evaluation, even though this improvement is not reflected in the results of automatic evaluation.

## 5.6 Performance on Large Language Models

We investigate the benefits of our task-adaptive tokenization on the effectiveness and efficiency generation for the recent LLMs. Tab 5 shows the results when using the 7B LLaMa model, as described in Sec 5.3. The RougeL score increases by 15.0% when applying our tokenization, which indicates that our task-adaptive tokenization brings about additional and model-agnostic performance benefits.

| Setting | Bleu | +pct | B-1 | B-3 | R-L | +pct |
|---|---|---|---|---|---|---|
| **CN PsyQA** | | | | | | |
| LLaMA$_{base}$ | 27.9 | - | 64.5 | 12.1 | 30.1 | - |
| LLaMA$_{TaT}$ | **29.8** | +6.8% | 69.6† | **12.5** | 34.0† | +13.0% |
| +mapping | 29.5 | +5.7% | **69.6**† | 12.3 | **34.6**† | +15.0% |

Table 5: Generation effectiveness on Chinese LLaMA. Bleu is calculated by averaging B-1,2,3,4, where B-n denotes the Bleu n-gram precision. R-L is RougeL score. +pct denotes the percentage of improving scores corresponding to Bleu and RougeL over the base. † indicates a significant improvement over the base (p-value < 0.05)

## 6 Further Analysis and Supplementary Experiment

We conduct supplementary analyses and experiments to gain deeper insights into the observed performance disparities between the Chinese and English datasets. Specifically, we analyze the constituents within the generated responses, and classify them into three categories to quantify their contribution to the overall generation length. These categories are as follows: (1) **overlap**, denoting tokens shared between the original and task-specific vocabularies; (2) **non-overlap**, representing tokens exclusive to the task-specific vocabulary; and (3) **original**, signifying tokens found solely within the original vocabulary. As exemplified in Tab 6, a noticeable discrepancy

emerges in the contribution of non-overlapped tokens when comparing the Reddit MHP and CN PsyQA datasets. Our hypothesis posits that the generation of newly-introduced task-specific tokens may play a pivotal role in explaining the less pronounced performance gains observed in the Reddit MHP dataset, in contrast to the CN PsyQA dataset.

| | overlap | non-overlap | original |
|---|---|---|---|
| CN PsyQA | 12.0% | 84.3% | 3.7% |
| MHP Reddit | 62.4% | 37.5% | 0.1% |

Table 6: Length contribution of three types of tokens generated on both datasets.

Further, we investigate whether the disparity in constitution analysis between the two datasets arises from linguistic distinctions or data quality concerns. As highlighted in the description of the datasets in Sec 5.1, participants in the MHP Reddit dataset primarily self-identify as professionals in their profiles. Additionally, we observe that many QA pairs in Reddit MHP resemble general chitchat, where task-specific terminology may be less prevalent. To address this, we translate 5,118 QA pairs from the CN PsyQA dataset into a parallel corpus, split into an 8:1:1 ratio (training, development, test). With this, we aim to re-assess the effectiveness of our proposed tokenization techniques within an English data context. As illustrated in Tab 7, task-adaptive tokenization markedly enhances generation effectiveness across all metrics. Based on these results, we conclude that our proposed tokenization method performs effectively in tasks involving frequent use of domain-specific expressions, compared to open domain communication.

| Setting | Bleu | +pct | B-1 | B-3 | R-L | +pct |
|---|---|---|---|---|---|---|
| **Translated PsyQA** | | | | | | |
| GPT$_{base}$ | 12.6 | - | 44.5 | **0.6** | 8.5 | - |
| GPT$_{TaT}$ | 19.5† | +54.8% | **58.1**† | 2.4† | **15.2**† | +78.8% |
| +mapping | **19.9**† | +57.9% | 56.3† | **4.0**† | 14.7† | +72.9% |

Table 7: Generation effectiveness on Translated PsyQA. See Tab 2 for column and notation definition.

## 7 Related Work

**Segmentation Algorithms and Vocabulary Development.** Mapping text into tokens, as a key step in the NLP pipeline, has a rich history of algorithms and vocabulary development. Early

proposals of segmenting text varied from utilizing linguistic cues at different levels (e.g., morpheme or syntax) to statistical methods (Creutz and Lagus, 2006; Luong et al., 2013; Zouhar et al., 2023). Notably, during the era of statistical machine translation, phrase-level translation, which shares a similar idea with variable segmentation, was one of the most promising translation methods at that time (Koehn et al., 2007, 2003). This paradigm enjoyed considerable popularity until the rise of deep learning techniques, shifting the focus to subword-level segmentation, given the need to address the challenge of poor representation of rare/sparse words (Sennrich et al., 2015, 2016; Kudo, 2018; Kudo and Richardson, 2018). This approach largely improves performance of NLP models by leveraging shared subword units and maintaining a compact vocabulary. In parallel, the use of vocabulary transformed with the advent of large language models (LLMs). Previously, each model tended to develop its own vocabulary in isolation (Jean et al., 2015), but recent work started to directly use the vocabulary of pre-trained models to inherit the strong representations acquired through pre-training. Despite existing work(Bagdasaryan et al., 2022; Liu et al., 2021d), customizing vocabularies for specific tasks lost popularity due to challenges in integrating them into nowadays pretraining-finetuning paradigm.

Recently, research has been proposed to for tokenization quality evaluation (Zouhar et al., 2023) and token cost analysis of a tokenizer among different languages (Ahia et al., 2023), indicating the researchers' increased concerns on tokenization improvement. It is worth noting that Liu et al. (2021d); Sachidananda et al. (2021) also addressed the vocabulary gap between pretraining and finetuning or domain-level post-pretraining; however, their solutions either requires an additional model module for token alignment or solely operates at the sub-word level. In contrast, our work provides a model-agnostic solution and embraces the merits of flexible variable segmentation cherished in earlier research while still retaining the ability to leverage existing pre-trained models.

**Generation in Mental Health Support.** In recent years, several studies have explored the application of generation techniques for mental health support (Shen et al., 2022b; Lee et al., 2021; Sabour et al., 2023; Hsu et al., 2023; Liu et al.,

2021b), including counseling-style dialog generation systems (Shen et al., 2020) and the incorporation of counseling strategies in response generation (Sun et al., 2021). Furthermore, recent work has investigated the use of large language models as expert systems for support strategy counseling (Zhang et al., 2023). However, rather than focusing adaptation effort on fine-tuning or prompting, our study focuses on tokenization, an easily overlooked component in the NLP pipeline. We hypothesize that for tasks in technical writing fields, e.g., mental health, adapting the tokenization to the downstream language style is a potential strategy to unlock additional performance.

**Text Generation Efficiency.** Recent years have witnessed a great surge of interest in enhancing text generation efficiency due to the need of long-form text generation or the trend of scaling up NLP models (Hessenthaler et al., 2022; Tay et al., 2020). Efforts on improving generation efficiency range from researching non-autoregressive modeling techniques (Li et al., 2022; Qi et al., 2021), attention mechanism optimization (Yan et al., 2021; Beltagy et al., 2020), model size reducing (Zafrir et al., 2019; Ma et al., 2022), and infrastructure innovations (Rasley et al., 2020; developers, 2021, 2022; Fang et al., 2021).

## 8 Conclusion

In this paper, we proposed task-specific tokenization as a way to adapt the generation pipeline to the specifics of a downstream task. We demonstrated the efficiency and improved long-form quality of generation for the domain of mental health, where we specifically addressed the task of psychological question answering. Our tokenizer leverages the specifics of the downstream task data, while still retaining the ability to integrate into existing pre-trained models. Through extensive experiments, we demonstrated the ability of task-adaptive tokenization to enhance both the effectiveness and efficiency of long-form generation.

We believe our work is particularly useful in the era of large language models (LLMs), as the proposed task-adaptive tokenizer can lead to significant improvements while being domain and model agnostic. Based on our findings, we suggest plug-and-play tokenization for LLMs when performing specific generation tasks.

## Limitations

Despite the strength of our proposed task-adaptive tokenization, several limitations remain. In particular, due to limited resource, we were only able to test it on one dataset and on a large-scale language model. Future work should consider evaluating the effectiveness of our task-adaptive tokenizer on additional domains and LLMs. The effectiveness of this tokenization should also be verified on additional languages and models of various sizes. Finally, in our experiments, we found our tokenization does not significantly enhance the generation speed in English, which may be due to the fact that the English vocabulary has less room to increase its granularity compared to a character-based language like Chinese.

## Ethics Statement

Several ethical situations should be considered. First, due to the black-box feature of neural networks, we do not recommend any generation technique including our proposed method to be directly used by a mental health supporter. Instead, in practice, it could be exposed to well-trained MHPs informed by the pros and cons of using generative AI, and offered as a suggesting tool for professionals. The professionals should be fully informed and trained to be accountable for the second-edited content. Second, during human evaluations, we informed all the human evaluators that the responses they would see might cause some discomfort, so they could decline their participation in the evaluation. Finally, regarding the use of the PsyQA data, we have received authorization to use the data and conduct research on it ethically and legally through approval from the data publishers.

## Acknowledgements

We acknowledge the valuable feedback from the reviewers, as well as from Orevaoghene Ahia, a Ph.D student at University of Washington and Vilém Zouhar, a Ph.D student at ETH Zürich during the discussion about paper improving for the camera-ready version. We thank the anonymous reviewers for their constructive feedback. We are also grateful to the members of the Language and Information Technologies lab at the University of Michigan for the insightful discussions during the early stage of the project. This project was partially funded by an award from the Templeton Foundation (#62256).

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

## A Few-shot Inadequacy in PsyQA task

**Limited Input Size and Inconsistent Results.** Due to length restrictions, using more than one PsyQA example in the prompt was not allowed for longer post-response pairs, making the few-shot in-context learning for this task impractical. In addition, in cases where we were able to provide more than one example as the prompt, the model generally produced low-quality incoherent responses that were inconsistent with the input context (i.e., user's post). We believe this is expected as the model would have difficulties distinguishing between the posts in the few-shot examples and the post that requires a generated response.

**Misalignment with language behavior of MHPs.** The language style of LLMs is adopted from their corresponding pre-training data and human feedback, which may not meet the requirement of linguistic behaviors of MHPs. In other words, despite the significant performance of recent LLMs, desirable performance could only be achieved through finetuning the model on professional data that captures the rich MHP language style and patterns.

## B Statistics on tokens at different lengths in a merged vocabulary

Table 8 presents the top-scored tokens at various different lengths to show that the task-adaptive tokenizer successfully includes many task-specific terminologies and expressions in its vocabulary.

**Post**

A 20-year-old male with bipolar disorder and symptoms of OCD, who is afraid of attractive women, is seeking treatment?

I have bipolar disorder and OCD symptoms, and I fear attractive women. I feel nervous and sweaty on the subway, and anxious when passing by attractive women on the street ... Psychoanalysis suggests fear of being seen through or assigning sexual meaning to interactions. I have experienced trauma from the opposite sex in college. I tried exposure therapy but it didn't work. Is hypnotherapy effective? Are there other treatment options like medication? I am anxious.

Keywords: diagnosis, treatment, counseling

**Response**

Hello! It is crucial to seek professional psychological therapy and counseling for your condition, as bipolar disorder with symptoms of obsessive-compulsive traits and opposite-sex fear is a severe mental health issue that requires proper evaluation and treatment by a qualified mental health professional. The fear of the opposite sex may stem from past traumatic experiences and involve issues related to self-disclosure, sexual desires, and identity transformation.

Cognitive-behavioral therapy (CBT) can be an effective approach for addressing opposite sex fear. CBT is a common therapeutic approach that focuses on identifying and changing irrational thoughts and behavior patterns to improve mental health. Strategies may include recognizing and understanding the fear, gradual exposure to the opposite sex, cognitive restructuring to change negative thought patterns, and learning relaxation techniques to manage anxiety.

....

Exposure therapy and hypnotherapy can also be effective treatment approaches. Exposure therapy involves gradual exposure to the fear object to reduce fear, while hypnotherapy uses hypnosis to modify thoughts and behaviors. However, it is important to undergo these therapies under the guidance of a qualified professional.

In addition to psychotherapy, medication can be a viable treatment option for bipolar disorder. Common medications used include lithium salts, antidepressants, and antipsychotics. However, medication should be prescribed and monitored by a qualified healthcare professional due to potential side effects.

It is crucial to seek professional mental health care and counseling promptly. Mental health issues require timely treatment to prevent further impact on well-being. Wishing you a speedy recovery.

Figure 3: An example of PsyQA.

## C Experiment Details

Except for the experiments on vocabulary size, we adopted a uniform size of task-specific vocabulary for task-adaptive tokenizer construction by merging a 10k task-specific vocabulary and the original vocabulary of each pre-trained model. During training, the regularization coefficient for sampling segmentation among various results was 0.5. We use one Nvidia GPU A40 for training GPT and Bart, which can load eight samples with a padding length of 1024. Training parameters are max_length=1024 for CN PsyQA/512 for MHP Reddit, batch_size=8, training_epoch=30, warmup_ratio=0.1. We use different decoding parameters for different models, as listed in Table 9.

| #length | top-5 tokens |
|---------|--------------|
| (0, 6] | ., ",", , s, _and |
| (6, 12] | _people, _really, _if_you, _things, _yourself |
| (12, 18] | _relationship, _mental_health, _relationships, _your_therapist, _that_you_are |
| (18, 24] | _It_sounds_like_you, _with_your_therapist, _in_the_first_place, _it_sounds_like_you, guide_to_conversation |
| (24, 30] | _a_mental_health_professional, omeone_will_do_the_right_thing, _When_trust_is_broken_one_of_t, _the_most_important_thing, _mental_health_professionals |
| (30, 32] | _borderline_personality_disorder, _community_mental_health_center, _Borderline_Personality_Disorder, _Licensed_Professional_Counselor, _understand_exactly_how_you_feel |

Table 8: Statistics of tokens with the highest log-likelihood scores in each length interval. The task-specific vocabulary for statistics is built on MHP datasets.

| Setting | Parameters |
|---------|------------|
| **CN PsyQA** | |
| GPT | top_k=50, top_p=0.9, do_sample=True, repetition_penalty=1.5, temperature=0.9 |
| Bart | top_k=50, top_p=0.9, do_sample=True, repetition_penalty=1.5, temperature=0.9 |
| LLaMA | top_k=40, top_p=0.8, do_sample=True, num_beam=4, repetition_penalty=1.2, temperature=0.95, max_new_tokens=512, no_repeat_ngram_size=4 |
| **MHP Reddit** | |
| GPT | top_k=50, top_p=0.9, do_sample=True, repetition_penalty=1.5, temperature=1 |
| Bart | top_k=50, top_p=0.9, do_sample=True, repetition_penalty=1.1, temperature=1 |

Table 9: Decoding Parameters

## D  Vocabulary size

We reported the generation effectiveness cross different task-specific vocabulary sizes in Table 10.

## E  Case Study

For each generation setting, we present one generated case. The results are in Table 11 and Table 12. From the result on MHP Reddit, we see due to the scarcity of the data, generation quality is hurt by poor representation of newly-add tokens while fluency, coherence, and professional expression are recovered after the application of the mapping mechanism.

| Taks-specific vocab size | Increment | Bleu | R-L |
|---|---|---|---|
| | **CN PsyQA** | | |
| **6k** | 2774 | 27.31 | 22.48 |
| **10k** | 6756 | 27.11 | 22.05 |
| **14k** | 10737 | 27.04 | 21.52 |
| **18k** | 14723 | 26.94 | 21.46 |
| | **MHP Reddit** | | |
| **6k** | 2367 | 4.81 | 10.72 |
| **10k** | 5470 | 4.77 | 10.71 |
| **14k** | 8278 | 4.78 | 10.76 |
| **18k** | 11733 | 3.92 | 9.81 |

Table 10: Comparison result on different sizes of specialized vocabulary. Especially, **taks-specific vocab size** is the size of vocabulary before merging. **increment** means the token increase relative to base vocabulary after merging the task-specific and base vocabulary into one. **Bleu** denotes the average of Bleu-1,2,3 and 4. **R-L** denotes Rouge-L.

Figure 4: Human evaluation guideline.

| Context |
|---|
| [QUE] 每份工作干不长，心一直空的，缺乏安全感，如何摆脱? [DESC] 工作了三四年，每份工作都干不长久。一旦熟悉工作，稳定下来了，每天干着重复的工作，一段时间后，每天就会觉得心里很浮躁很不安。感觉生活像一潭死水，想着有很长一段时间要这样过，心里就难受，害怕。待的时间越久，就有些想自残，甚至产生轻生的念头。明明工作上没什么问题，老板都是认可的，每次跟他们辞职，他们也不想我走，都会劝着我留下。虽然自己不爱说话，但因为长得不丑，又有点乖巧，同事刚开始也愿意主动接近我。发现我不爱说话，就有些疏离，但真正有事，自己外貌看上去可怜，他们少数又会主动帮我。开始每份工作都做了半年才辞，后来的做到了三四个月就有些奔溃。有一次没什么事，自己情绪很糟就哭了，别人问我也无法解释。有一份坚持了一年多，之后辞了就很想轻生。我一直都不明白活着是为了什么，心一直都是空的。有男友了也从不信他们真的爱我，一直也觉得我妈不爱我。一直觉得自己和这个世界隔着什么，总觉得自己人生无望，活着没盼头[KWD] 成长,性格完善,工作学习,人生意义 |

| Genrated Response | | |
|---|---|---|
| **M** | **NM** | **B** |
| [QUE]楼主你好!看完你的文字描述先给你一个温暖的拥抱 从你的描述中可以看出你是一个非常善良、温柔的女孩子呢!!!无论做什么事情都会有"怕"的心态。但是在现实社会里却可能有这样的想法——如果把人作为一个具体事件来看待的话确实很难实现这个要求。其实你的内心对于这件我们认识不全面的事会有很多疑惑的想法和情绪，所以会觉得总是活在自己自己的想象中,并且会有一点抵触的感觉:一方面是你觉得现在这份工作已经不是自己想要做的样子;二是还有一种你还没有真正的喜欢它......好像没有这么坚定的信念而犹豫徘徊很久。【1)首先要明白清楚自己想要什么职业所在。你想从事什么样的专业事业是可以成为你想要的样子的人吗?(比如你可以学些什么技术等等),这是必须要一步一步地去实践才能完成的事情。2.其次需要了解自己在当下的岗位上遇到的一些困难或者遭遇一些挫折是什么导致你现在出现这种纠结的感受的原因。(比如说说换工作这件事让你很难受等)。这里涉及到很多信息哦)接下来就是解决困惑的关键步骤。1.找到你喜欢做的事情并付诸行动。3.想清楚你自己的目标方向在哪里3.弄清楚自己到底想要的是什么以后再去做选择。4.找出自己的兴趣点最后就是要找到自己喜欢做的事与热爱。当你将你想要做的定义出来的时候,就会看到更广阔的世界啦(当然如果你确定一定要去做自己喜欢的行业那也是可以呀。)世界和我爱着你, 祝好 。 | 题主你好 看到你的描述了解到题主的烦恼是被职场性的孩子困扰的,希望父母能够理解你的感受呢!我想她和你成长经历有关:这让你产生了很大的困扰,所以可能对父母的要求太高的要求可能是更喜欢、去掌控你的优点。【梳理反馈】首先自我暴露的过程需要学会慢慢积累自己的情绪感受,而不能接受自己的负面情绪。通过外在方式与控制的情绪宣泄出来,当它无法承受或者失望的时候证明自己就会更加强烈愤怒。2关于社交方面的事情,我们是否感到迷茫时出现这种心情反应呢?2.如果这些压力也不太一样的话楼主或许有时候会出现一些无力无助的现象并且导致在逃避的原因中看到,其实也可以看看这些问题是什么,比如今天考研没有动力吗等等事情好,似乎只能继续让自己让家人朋友轻松很多。这些都是为了具体计划的方法才能解决这样的事件。3你说自己总是陷入"不好"上突然消失之地",那么就会把目标当成一个目标进行积极能量的方法,同时不要过度自信,而且他也不是努力的人。然后认真思考一下这一点对方有什么代价感不同的建议对你造成了一定的影响导致的。你可以做一些运动不断的提升训练的地方2.不必强迫自己做冥想练习一下注意力转移到其他同学身上的压力当中寻找专业的咨询师的心理咨询师倾诉。(2)其次及时锻炼休息的时间的时间。2、学会放下手机的变化等人聊天。 | 你好呀 非常理解您的困扰，抱抱楼主～首先我身边还存在着许多[UNK]懒惰症[UNK]患者(表面特容易改变)其实我建议你：1.学会拒绝他人不足部分原本并未知道怎样与环境相处、以及与他人交流等行为只能通过向周围比较亲密或者让他人舒服的方式进入对象关系中获取满足，当达到某种程度时再转移注意力2.找出适合应用于自卑内驱者和焦虑抑郁症群体间反差最大的心结。3.寻求专业咨询师的支援尽管目前社会存在各类障碍需引起更加强烈地重视但是希望我所提供的分析仅代表个人观点，谢谢！ |

Table 11: Case study on CN PsyQA. An explanation for abbreviations: M for GPT2$_{TaT}$ +mapping, B for GPT2$_{base}$, and NM for GPT2$_{TaT}$ w/o mapping;

| | Context | |
|---|---|---|
| NM | B | |

Context

[QUE] can we talk about dissociation?<|sep|>[DESC] i'm just wondering what other people's experiences with dissociation are, and if there are different types? personally, my way to dissociate is kind of an attempt to evoke/"be" stronger versions of myself (always male versions though - which i find interesting - i've no desire to be male irl, just in my head) when i'm under duress of some sort, usually pain or anxiety. is this unusual? my boyfriend, on the other hand, calls it dissociation when he kind of "zones out", he says his eyes glaze over and he stops doing anything or really thinking anything (also in times of stress). are we both correct in calling what we experience "dissociation"? what's your experience? i understand this is kind of a taboo/sensitive topic, i'm just now getting to the point where i can openly talk about it, but i'm desperately curious about it. edit: thanks everyone for sharing, it's a fascinating subject. it's sounding to me like what i experience is different from "normal" dissociation somehow, and i'm going to investigate this further. <|sep|>[KWD] bpd<|sep|>

| Genrated Response | | |
|---|---|---|
| M | NM | B |
| if you felt as if you were experiencing opposed personality types, then maybe it's important to tell your friends about this because this might be useful to help you feel more confident and connected with life. please also know that some people experience their identity through distinct body parts and experiences, and that their bodies are different from what you described. | that also work for a onlyt up so some of my relationship-cunllalerded before she told me how but was all the feeling more it is get them out when they say after not him to do me! dont know has been will be if you need her off something as a being able to your first one if i am to get do i do because there are time you never want people who start again in the have been had as i was gonna about around... i always felt i had, npoxges has done than i know friends, but it can help feel nothing that you since. and there will be on my own life back that got help down without someone in farinilance; i feel even a lot what to say byasment cyst loves me better. | if you say so yourself then perhaps that could shed light into something deeper |

Table 12: Case study on MHP Reddit. An explanation for abbreviations: M for GPT2$_{TaT}$ +mapping, B for GPT2$_{base}$, and NM for GPT2$_{TaT}$ w/o mapping;