# OpenReview forum: "Task-Adaptive Tokenization: Enhancing Long-Form Text Generation Efficacy in Mental Health and Beyond"
_EMNLP/2023/Conference — EMNLP 2023 Main_

### Official Review · Reviewer_xPLS · 2023-08-03

**Soundness:** 4

**Excitement:**

4: Strong: This paper deepens the understanding of some phenomenon or lowers the barriers to an existing research direction.

**Paper Topic And Main Contributions:**

This paper introduces the novel strategy of task-adaptive tokenization as a method to adapt generation pipelines to specific downstream tasks and applies it to long-form generation in the domain of psychological question-answering tasks in both Chinese and English languages.


**Reasons To Accept:**

The paper addresses the prevalent problem of downstream domain adaptation of foundational language models through a new lens that goes beyond the common approach of post-training fine-tuning to propose a strategy of task-adaptive tokenization. This new strategy proposes building a tokenizer from the task data with variable segmentation and introduces a method for integrating with pre-existing training tokenization to seamlessly extend to off-the-shelf language models.
The strategy has the potential to be applicable across multiple downstream domains and across languages – showing success in both character and word-based languages in included experiments.
The paper is very clearly written, easy to follow and understand as a reader with the motivation, methodology, and experiments explained in detail.


**Reasons To Reject:**

While the paper provides two evaluation methods (automatic and human) for the main proposed task-adaptive tokenization strategy, further quantification of the results for the proposed strategy of token mapping is suggested. For example, from the results of human evaluation analysis, the paper claims that the use of the token mapping strategy for non-overlapping tokens is “crucial to ensure a robust token representation system”. Further quantification of the change in frequency of response tokens for original, overlapping, and non-overlapping tokens is suggested to provide more background to the benefits of employing the mapping strategy relative to model generated responses.


**Reproducibility:**

5: Could easily reproduce the results.

**Reviewer Confidence:**

3: Pretty sure, but there's a chance I missed something. Although I have a good feel for this area in general, I did not carefully check the paper's details, e.g., the math, experimental design, or novelty.

**Typos Grammar Style And Presentation Improvements:**

Suggestion to address drop in performance for non-mapping task-adaptation tokenization in MHP dataset in human evaluation section. More explanation in regards to performance drop due to MHP’s scarcity of data is offered in Appendix E but would benefit to also be addressed in the human evaluation section as well if possible.

---

> ### Author Rebuttal · Authors · 2023-08-27
>
> Thank you for your time and consideration in reviewing our work. We are grateful for your positive comments regarding the applicability of our approach, and the clarity of the motivation and methodology.
>
> To answer your comments:
>
> 1. Regarding the quantification results for the token-mapping mechanism.
>
> From the human evaluation and automatic evaluation, we found that
>
> (1) With smaller datasets, it becomes challenging to ensure adequate sampling for the new tokens. We can leverage the mapping mechanism to use the learned representations from the sub-tokens to initialize these new tokens. This would also considerably guarantee the model’s representation ability of new tokens, which explains the significant performance drop on fluency and local consistency, which points to a defeat in foundational representation ability, in human evaluation when mapping is absent.
>
> (2) With larger datasets, we found the mapping mechanism continued to bring  benefits. In general, we found that leveraging pretrained token knowledge is always helpful. As seen in the human evaluation, when the mapping mechanism is absent, the performance on fluency and logical consistency is not that bad compared to the results on scarce data, but still causes a decrease. However, mapping mechanisms may cause less task-specific terminologies than without mapping, which reflect a drop in automatic evaluation (mainly RougeL).  This is due to the increased possibility of sampling out pre-trained tokens influenced by mapping mechanisms.
>
> Hence, this is a tradeoff that should be decided depending on the application and dataset size. While we leave improvements in the mapping mechanism to future work, we will include statistics for change in frequency (based on your suggestion) in the final version of our paper.
>
> 2. Regarding the performance drop of non-mapping mechanism in human evaluation on the MHP dataset.
>
> With smaller datasets, it becomes challenging to ensure adequate sampling for the new tokens, which explains the significant performance drop in fluency and local consistency in human evaluation when mapping is absent. One solution is what we proposed, namely to use a mapping mechanism to alleviate this problem. Another possible solution is to resample at every training epoch, which can be  a trick to enhance the chance for each new token to be sampled out. We will add this experiment in the camera-ready version, and also include these additional clarifications.

---

### Official Review · Reviewer_iXkS · 2023-08-03

**Soundness:** 3

**Excitement:**

4: Strong: This paper deepens the understanding of some phenomenon or lowers the barriers to an existing research direction.

**Paper Topic And Main Contributions:**

The paper presents a task-adaptive tokenizer for better performance on downstream text generation task. The proposed method utilizes task-specific data to construct a vocabulary with different segmentation granularity, which can be merged into pretrained model.

**Reasons To Accept:**

1) The motivations of the proposed method is supported by cognitive linguistic studies.
2) The proposed method is proven to be effective and efficient on different pretrained models.

**Reasons To Reject:**

1) More examples could be included in the case study to highlight how the mapping mechanism improves model performance.

**Reproducibility:**

3: Could reproduce the results with some difficulty. The settings of parameters are underspecified or subjectively determined; the training/evaluation data are not widely available.

**Reviewer Confidence:**

2: Willing to defend my evaluation, but it is fairly likely that I missed some details, didn't understand some central points, or can't be sure about the novelty of the work.

---

> ### Author Rebuttal · Authors · 2023-08-27
>
> Thank you for your time and consideration in reviewing our work. We are glad to see you found our method to be effective and efficient, and you appreciated the cognitive science foundations of our work.
>
> To answer your comments:
>
> 1. About the need for more case studies.
>
>     We limited the number of case studies because of the space, but we agree that more case studies will increase the strength of our argument. Given the additional page allowed for the camera-ready, we will plan to include case studies in main body and additional cases in appendix.

---

### Official Review · Reviewer_m6qr · 2023-08-04

**Typos Grammar Style And Presentation Improvements:** None.
**Soundness:** 4

**Excitement:**

3: Ambivalent: It has merits (e.g., it reports state-of-the-art results, the idea is nice), but there are key weaknesses (e.g., it describes incremental work), and it can significantly benefit from another round of revision. However, I won't object to accepting it if my co-reviewers champion it.

**Missing References:**

None.

**Paper Topic And Main Contributions:**

This paper studies how to adapt the existing tokenizer to some domains that have many special terminologies. They propose to generate a new task-specific vocabulary. By merging the new tokens with the existing one and assigning sample scores, one sentence can be tokenized as different outcomes. They also study if better initialization affects the performance. Experimental results show the effectiveness of the proposed method.

**Questions For The Authors:**

- For one training example, how many instances will you sample to train the task-dependent model? If it's larger than one, it's hard to tell the improvement comes from better tokenization or larger number of training examples. I suggest that testing with only using the newly constructed vocabulary, this can be served as a very important baselines to verify if the constructed vocabulary is necessary.
- I am wondering if the proposed method works for the general tasksNon other than mental health related tasks. I feel that the proposed method is kind of "tokenization ensemble" so it should work for the general tasks.

**Reasons To Accept:**

- An interesting way to handle domain mismatch and task-dependent terminologies.
- The proposed method is simple and easy to implement.

**Reasons To Reject:**

- Sampling would increase the number of training examples. Given the current experiments, it's hard to tell the improvement comes from better tokenization or larger number of training examples.

**Reproducibility:**

4: Could mostly reproduce the results, but there may be some variation because of sample variance or minor variations in their interpretation of the protocol or method.

**Reviewer Confidence:**

4: Quite sure. I tried to check the important points carefully. It's unlikely, though conceivable, that I missed something that should affect my ratings.

---

> ### Author Rebuttal · Authors · 2023-08-27
>
> Thank you for your time and consideration in reviewing our work. We are glad you found our approach interesting and you appreciated the ease of implementation of our proposed method.
>
> 1. Regarding the change in the number of training instances
>
>     We would like to clarify that sampling **will not** increase the number of training examples; while it’s true that the number of tokens in the training data may change, the source training data does not change.
>
>     In our experiments, we use the same number of training samples regardless of the sampling method. The same phrase shown in different training examples may be tokenized differently (see Figure 1), but one training example will still only point to one tokenization result, whether by sampling or by mapping. Hence, the number of training instances does not change.
>
>     We do acknowledge that we should improve our writing so that this point is more clear, thank you for pointing this out.
>
> 2. Concerning the generalization of our approach
>
>     Our proposed approach is indeed a new approach to tokenization that can be generally applicable to generation tasks. As a first step,  we chose to demonstrate our approach on one domain (Mental health), so that we can perform extensive analyses and comparisons while keeping the vocabulary consistent. In the future, we plan to test the effectiveness of our approach on other domains (e.g. medical texts), and also adjust our approach so that it can be used in a  plug-and-play manner for current  LLMs.

---

### Meta-Review · Area_Chair_sZw8 · 2023-09-20

**Recommendation:** 4

**Metareview:**

The paper proposes a task-adaptive tokenization method and shows its efficacy on long-form generation in mental health. The method is simple and easy to integrate with the potential to be applicable across several downstream tasks and languages. The paper can be further strengthened by adding results on other tasks and by adding the analysis as provided in the rebuttal.

---

### Decision · Program_Chairs · 2023-10-07

**Decision:**

Accept-Main

**Comment:**

The paper proposes a task-adaptive tokenization method and shows its efficacy on long-form generation in mental health. The method is simple and easy to integrate with the potential to be applicable across several downstream tasks and languages. The paper can be further strengthened by adding results on other tasks and by adding the analysis as provided in the rebuttal.